



# Potential of load and O&M costs reductions of Multi Rotor System for the south Baltic Sea

Maciej Karczewski[1], Piotr Domagalski[1], Arnoldus van Wingerde[2], Bernahard Stoevesandt[2], Peter Jamieson[3], Lars Roar Saetran[4]

[1]WindTAK sp z o.o. [Ltd], Wroblewskiego 38A, 93-578 Lodz, Institute of Turbomachinery, Lodz University of Technology, Wolczanska 219/223, Lodz 90-924, Poland

[2] IWES Fraunhofer, am Seedeich 45, 27572 Bremerhaven, Germany

[3] University of Strathclyde, Royal College Building, 204 George Street, Glasgow G1 1XW, United Kingdom.

[4] Department of Energy and Process Engineering, NTNU Trondheim, Kolbjørn Hejes vei 2, 7491 Trondheim, Norway

*Correspondence to*: Maciej Karczewski (maciej.karczewski@windtak.pl)

**Abstract**. Many coastal regions in Norway, Spain, Portugal, Japan or the United States are comprised of large water depths (>50m) making the installation of typical bottom-fixed off-shore wind turbines very difficult and expensive. This is the reason why the floating wind turbines (FOWT) are a promising solution able to exploit the high energy potential contained in these regions. The Advanced Multi-Rotor Turbine for Deep Water Off-shore Energy (AMRowe) has been undertaken to design and develop a cost–competitive, innovative floating Polish multi rotor system, aiming at the optimal usage of European off-shore wind potential. In the article, a prospective deep off-shore location in the south Baltic Sea is identified. The authors built a cost model to check its suitability relative to the sites already commissioned by the Polish government. A set of metocean conditions tallied for a 50-year period is then used to assess performance of the proposed multi rotor floating wind turbine and to benchmark it against a single rotor 5 MW baseline turbine. The typical load cases are also investigated to observe impact on a single rotor blade in an multi-rotor arrangement in order to begin search for the key design drivers.

## 1 Introduction

The current technology drive in regards to off-shore floating wind turbines requires a development of innovative control strategies and substructure concepts. One of the main objectives is minimalisation of wind and wave-induced platform motions to provide a stable operational environment to the wind turbine rotor. The second group of objectives targets mass as well as installation, operational and maintenance (IO&M) costs reduction.

The present prototypes of wind turbines operating at deep sea are classical three-bladed wind turbines. These turbines are known for their superior efficiency and performance for on-land based applications. In fact, at present, on land and in shallow sea waters, a 3-bladed HAWT is able to generate up to 12 MW of power (e.g. GE Haliade-Xa ). This is a direct result of increasing the rotor size. However, with an increased rotor size, the generator mass, and hence the nacelle it will reach gigantic dimensions. The so-called Top Head Mass (THM) of the Rotor Nacelle Assembly (RNA) can reach 500 tonnes (e.g. Vestas V-164). An application of wind turbine at deep sea (over 50m+ depth) requires a modified approach: a turbine that is able to float on the surface of the water. Technologically, by further upscaling of turbines, one runs the risk of increased forces and moments acting on the RNA at substantial heights above the water surface. On the cost side, large RNAs create problems in terms of transportation (towing to an operational site) and installation by dedicated vessels and cranes. Hence, a new approach is suggested: a floating advanced multi rotor system turbine. By combining a few smaller designs it might be possible to decrease the IO&M costs simplifying the manufacturing, towing, and maintenance processes, while being able to keep competitive or even superior energy efficiency.





## 2 Cost Analysis

A number of tools were used to assess the sea conditions and legislative status of energy harvesting on Polish territorial waters of the Baltic Sea extending up to 200 km north of the Gdansk port and west in the direction of the city of Szczecin.

The ORECCA project (Off-shore Renewable Energy Conversion platforms – Coordination Action) helped to establish the water depth levels and average wind speeds, while documents (Gutkowski et al., 2012, Ministry of Economy of Poland, 2010, Zaucha and Matczak, 2017) were used to name 3 prospective single rotor wind turbine (WT) locations: one on-shore at the sea-side (1), which would be the reference location for this study, and two bottom-fixed shallow off-shore (2 and 3). Based on the complex map prepared by the authors of the article who combined the water depth levels, average wind speeds,

animal protected zones, the layout of the existing HVDC lines and marine vessel routes (available on marinetraffic.com), the 4th location is proposed in a deep off-shore sea about 100 km north of the Polish coastline (figure 1). The National Renewable Energy Laboratory (NREL) US-based cost scaling model (Fingersh et al. 2006) was employed to assess the overall costs of the investments in all 4 locations. All costs were recalculated into PLN basing on the annual mean currency rate of USD-PLN from the year 2003 on. The rates published by the National Bank of Poland were used and corrected for

inflation, year-by-year, according to historical data from Central Statistical Office of Poland. Because a detailed due diligence fact sheet for a Polish on-shore location was made available to us, the benchmark turbine for cost analysis was Vestas V100 2MW, except that for the location 4 we implement a floating multi-rotor system (MRS) of the equivalent swept area and nameplate capacity of 2MW. The reader is advised that an average water depth at the site 4 is around 60-100 m. Moreover, the cost model for location 4 was enriched by findings from a study of Stehly et al. 2017, where a floating wind

turbine price breakdown was presented.

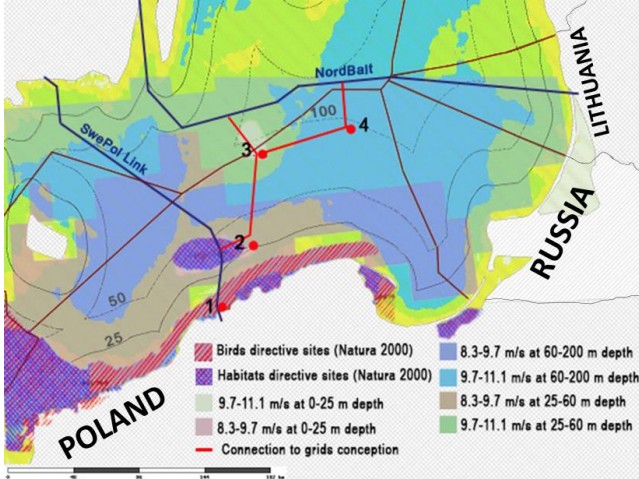

**Figure 1: Positions of analysed wind farms: on-shore (1), shallow off-shore (2/3) as commissioned by Polish gov't, and location proposed by the authors (loc. 4).**

### 2.1 Wind conditions on Baltic Sea

The wind data for cost analysis at locations 2 and 3 was obtained from prognostic calculations of the Unified Model v.6.2 performed by (Centre for Mathematical and Computational Modelling in Warsaw, 2012), for location 1 from (Kroszczynski K., 2012). The first set was assessed over the period of one year in 2012 and the second for 2013, both at the height of 10m above mean sea level (AMSL). We recomputed it for the 100m AMSL using the power law with exponent equal to 1/7 (Germanischer Lloyd WindEnergie, 2005). In absence of wind data for location 4, we used the numbers from location 3

assuming no differences. Table 1 states the Weibull function parameters for all sites.



**Table 1: Average wind speeds at 100m AMSL.**

| Loco | $V_{avg}$ at 100m AMSL (m/s) | $k$ shape factor | $A$ scale factor |
|------|------|------|------|
| **1** | 7.36 | 2.18 | 8.31 |
| **2** | 10.35 | 2.26 | 11.69 |
| **3,4** | 11.36 | 2.54 | 12.80 |

In order to estimate the production potential at all four sites real production data was used as a reference. (Staporek and Tauzowski, 2017) in a DNV GL report on wind energy converters operating in the southern and the northern Poland, state the average production ratio (6000 GWh, or the net annual energy production $AEP_{net1}$=2996 MWh/MW/yr) and the average

capacity factor ($CF_1$=0.342) for the Vestas V100-2MW units situated in the north, where the reference turbine nr 1 is located. This data was used for calculations in our cost model. For locations 2 and 3 an extensive production data set for the bottom-fixed off-shore projects Horns Rev I and II was used (Danish Energy Agency, 2020). We post-processed raw data and prepared figure 2 to present the average yearly capacity factors of the three Horns Rev offshore wind farms.

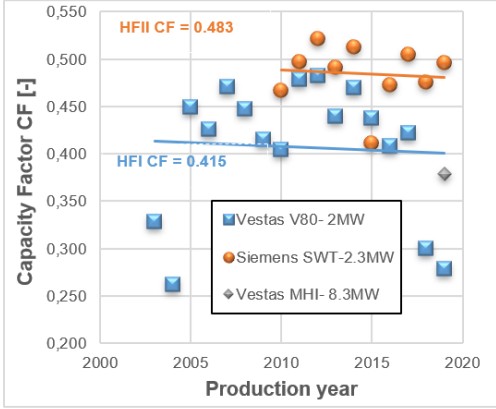

**Figure 2. Annual Energy Production (AEP) data for the bottom-fixed off-shore wind farm Horns Rev I, II and III (HFI, HFII, HFIII) – own plot based on Danish Energy Agency data**

The HFI consists of Vestas V80-2MW turbines and is located about 30 km from the west coast of Denmark. Therefore, in this study $CF_2$=0.415 ($AEP_{net2}$=3635 MWh/MW/yr) was used as the average capacity factor for the location 2. The more distant HFII situated about 60 km from the coastline, where Siemens SWT-2.3 MW turbines are used, achieved the average

CF of 0.483. This value was utilized for a more distant location 3 in our study ($CF_3$=0.483 and $AEP_{net3}$=3679 MWh/MW/yr). In a comparative analysis made by (Chasapogiannis et al, 2014) it was shown that for the same wind conditions a 7-rotor MRS outperforms a single rotor turbine of the same nominal power by +2.6%. Therefore, for the site no. 4 we employed the $AEP_{net4}$=3775 MWh/MW/yr resulting in the $CF_4$=0.496. For comparison, the average capacity factor reported for the first FOWT project, the HyWind Scotland, amounts to 0.538 after 2 years of operation (Smith, 2020). Interestingly enough, in the

recent year the new HRIII, where the largest Vestas MHI-8.3MW turbines are in use, achieved a lower capacity than the older HFII (see figure 2).





**2.2 Cost breakdown results**

The total costs for sites 1, 2, 3 and 4 were divided into two main categories: ICC – initial capital costs (the turbine capital cost TCC, project, erection and energy development expenditures combined in the balance of station cost, BOS, as well as

financial costs: insurance, securing a loan for construction works, a site decommissioning bond and a contingency fund to secure a financial liquidity), as well as annual O&M charges, including site lease, accounting, telecommunication, the plant's own energy needs and overall wind turbine servicing expenditures.

The price of the turbine with transformer was indifferent for all locations, except location 4, where the MRS turbine costs are adjusted. The project, the erection and energy development expenses as well as the O&M were comparable for the off-shore

turbines 2 and 3. The differences came in the transportation monies, because the estimated distances between locations 2 and 3 to the port of Gdansk, currently planned for servicing Polish off-shore wind parks, amount to 82 and 92 nautical miles respectively. The off-shore towing cost was based on the expertise of Gdansk Shipyard LLC, which claims the marine transportation on the Baltic Sea to vary between €80 000 up to €250 000 depending on the size of the ship, the mass of the turbine parts and the distance to be covered. Based on the NREL cost model the transportation fees amounted to around

€105 000, a figure quite in the range above. The second major difference is the estimated distance that separates the electrical substations 2 and 3 from the nearest HVDC lines (see figure 1). This distance scales the cost of the electrical substructure 3 by the factor of 1.5. Furthermore, there are major cost differences between locations 1 and 2/3: the off-shore turbines undergo marinization works, what increases their TCCs, the BOS in off-shore projects is significantly higher due to the need for specialised vessels, expert-level personnel and high-end parts & materials, the financial costs increase as

investors must mitigate the risks of the open sea operations.

The rates of the 4[th] location are different due to the proposed turbine type, the floating MRS. (Jamieson and Branney, 2012) basing on the "square-cube" relationship (an overall MRS rotor mass can be reduced by $\frac{1}{\sqrt{n}}$ as compared to a single rotor of equivalent swept area, where n is the number of rotors) estimated the cost impact on the manufacturing expenses per turbine component. It was shown that the TCC of an MRS, having the identical nameplate capacity as a large single rotor, is reduced

by -61%. We thus scaled the TCC cost for location 4 by exactly the same figure, but kept the pricing of the support structure identical to locations 2 and 3. Moreover, the BOS at the site no. 4 is impacted since the foundation is replaced by the mooring structure and no scour protection is required. In terms of mooring, we assumed the system to be the same as for the Hywind Demo launched by Equinor for which the catenary mooring consisted of 3 lines of an approximately 800 m each (Neuenkirchen, 2013) with prices taken from (Skaare et al., 2011). Furthermore, we suggest that the increased BOS expenses

associated with the distant off-shore turbine nr 4 may be further offset by assembling MRS in the docks in order to tow the readied system to the operational site rather than employ expensive construction vessels. (Jamieson, 2016) reports a reduction of 20% of costs in this case, a figure reflected in our model. The turbine nr 4 is exactly the same distance away from Gdansk shipyard and from the nearest HVDC line as the turbine nr 3 (see figure 1). Hence, the electrical cabling cost and the transportation costs are the same.

On the other hand, the capital costs in case of floating wind turbines are likely to be higher as it often is with new pioneering technologies. Based on the HyWind turbines, financial needs of implementing the MRS at location 4 are computed after (Stehly et al., 2017), who suggest scaling factors to derive amounts for construction financing, contingency fund and decommissioning bond of a FOWT project. Finally, the costs of servicing the MRS are likely to be lower than for a large single rotor unit. As suggested in (InnWind, 2015) MRS can be equipped with an overhead crane system, for example as the

one depicted in (Jamieson, 2019). A 13% reduction in the O&M costs was reported in (Jamieson, 2016) by the avoidance of jack-up vessels for any level of open-sea rotor system failure. Our model for site 4 reflects this very reduction. At the same time turbines 2 and 3 would require on-site assembly and a more expensive fleet to maintain asset availability at high level.




This naturally elevates the O&M costs. Table 2 summarizes all calculations. To compute the levelized cost of energy, we used a fixed charge rate of 11.58% for all sites.

**Table 2: Repartition of the CapEx, OpEx and LCOE for wind turbine locations 1, 2, 3 and 4.**

|  | Vestas V100-2MW | | | MRS | Vestas V100-2MW | | | MRS |
| --- | --- | --- | --- | --- | --- | --- | --- | --- |
|  | Loc 1 (€/kW) | Loc 2 (€/kW) | Loc 3 (€/kW) | Loc 4 (€/kW) | Loc 1 (€/MWh) | Loc 2 (€/MWh) | Loc 3 (€/MWh) | Loc 4 (€/MWh) |
| **Turbine Capital Cost (TCC)** | **1269** | **1440** | **1440** | **557** | **49.1** | **45.9** | **45.3** | **17.1** |
| Development cost |  | 45.2 | 45.2 | 45.2 |  | 1.44 | 1.42 | 1.39 |
| Port and Staging |  | 24.4 | 24.4 | 24.4 |  | 0.78 | 0.77 | 0.75 |
| Support Structure |  | 367 | 367 | 367 |  | 11.7 | 11.5 | 11.2 |
| Electrical Infrastructure |  | 213 | 318 | 318 |  | 6.78 | 10.0 | 9.75 |
| Assembly & Installation |  | 273 | 279 | 187 |  | 8.71 | 8.77 | 5.73 |
| **Balance of Station (BOS)** | **360** | **923** | **1033** | **941** | **13.9** | **29.4** | **32.5** | **28.9** |
| Insurance |  | 190 | 190 | 73.6 |  | 6.06 | 5.99 | 2.26 |
| Construction financing |  | 70.9 | 74.2 | 410 |  | 2.26 | 2.34 | 12.6 |
| Decommissioning bond |  | 222 | 222 | 76.0 |  | 7.07 | 6.99 | 2.33 |
| Contingency |  | 247 | 247 | 447 |  | 7.88 | 7.78 | 13.7 |
| **Financial cost** | **119** | **731** | **734** | **1007** | **4.60** | **23.3** | **23.1** | **30.9** |
| **CapEx total** | 1748 | 3093 | 3207 | 2505 | 67.6 | 98.5 | 100.9 | 76.8 |
| **Total O&M** | **22.3** | **78.2** | **85.1** | **77.1** | **7.44** | **21.5** | **23.1** | **20.4** |
| **LCOE** |  |  |  |  | 75.0 | 120.0 | 124.1 | 97.3 |

The final expenditures are also presented in a pie-chart form in figure 3.

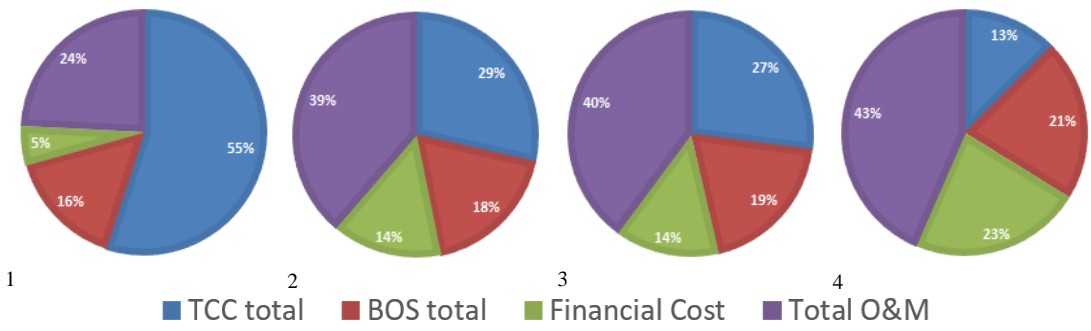

**Figure 3: Cost breakdowns for the analysed wind turbine sites.**

The plots reflect a typical CapEx and OpEx shares in the cost structure for the on-shore project, where the majority of

expenses is associated with the power plant cost and for the bottom fixed off-shore projects that see increased portion of the O&M expenditures. The information for location 4 is somewhat new and thus can serve as an interesting source of data for comparison. First of all, the TCC stake is further reduced – a tendency that is observable in many industries when technological advancements are made. However, the MRS are said to extend the limit of the maximum turbine capacity to 50 MW in a single unit what, given the historical LCOE reduction that was made possible by the development of the ever larger

single rotor turbines, suggests that future cost decreases in the TCC are definitely possible. Furthermore, the BOS and the O&M cost segments of the site no. 4 are just a margin away from other off-shore sites, +2-3% and +3-4% respectively to



locations 2 and 3. The biggest increase is observed in the financial part, estimated at +9%. This last observation may be treated as another good prognostic ahead of the MRS technology. Once the FOWT projects become more commonplace a further cutbacks of the capital cost is inevitable consequently reducing the overall LCOE of the MRS solutions even further.

**2.3 Cost of Energy comparison**

The key factor allowing the comparison of chosen locations is the cost of energy (COE). The COE formula was based on the NREL scaling model Fingersh et al. 2006 with turbine availability factor equal to 0.98 for all four sites. Table 3 combines the total costs and the LCOE.

**Table 3: Total costs, Annual Energy Production and Cost of Energy for the considered locations.**

|  | Location 1 | Location 2 | Location 3 | Location 4 |
|---|---|---|---|---|
| **Costs (mln PLN)** | 19.364 | 42.402 | 44.801 | 37.231 |
| **Costs (mln €)** | 4.610 | 10.096 | 10.667 | 8.865 |
| **Net AEP (MWh/MW/yr)** | 2996 | 3635 | 3679 | 3775 |
| **LCOE (PLN/MWh)** | 315 | 504 | 521 | 408 |
| **LCOE (€/MWh)** | 75.0 | 120 | 124 | 97.3 |
| **Relative to on-shore ref. (%)** | 0 | +60 | +65 | +30 |
| **Relative to location 2 (%)** | - | 0 | +3 | -18 |

The lowest COE ratio occurred for the reference on-shore power plant no. 1. The cost of erecting wind farms off-shore goes up by at least 60% in comparison to land-based installations. Furthermore, optimisation of O&M expenses can translate to very concrete savings (compare site 2 and 3), an observation quite consistent with the benchmark data found in (Maples et al., 2013). Despite the superior AEP, location 3 produced higher costs per energy output than the 2nd one, the effect of the pricier electrical cable lines, more expensive transportation that additionally re-appears as a recurring cost in the year-to-year
turbine servicing operations (increased O&M). In fact, the least favourable coefficient was calculated for the 3rd location. Compared to a standalone benchmark turbine, by economy of scale, the unit costs will be lower should the estimate be done farm-wise and for the modern turbines of much larger rated power. Nevertheless, the authors unanimously believe that the costs might scale down across all locations in such a case, so the conclusions would likely be similar.

Despite indisputable incentives of location 1, the publicly-owned electricity producers are going to be more pressed to erect
the off-shore turbines given the energy demand of northern Poland and the current legal status significantly curbing nearly the entire on-shore wind of Poland. After all, in locations 2/3/4 the AEP is well over 20% of that in the location 1. So, while capitalizing on the government plans to invest in shallow off-shore first, our results show it is worthwhile to consider the deep off-shore sites from the very beginning. For shallow off-shore the costs can be lowered by, for example, scaling the turbine up. However, this approach may eventually hit a technological and economic barriers before reaching 20MW rated
power in a single rotor machine. Thus, we propose a concept of a multi-rotor system (MRS) for location 4 to leverage the increased costs, the waive technological limit and the exploitation problems from the start. The cost simulation prepared clearly shows that the financial aspects do support such move as the floating MRS may be able to reduce the LCOE by up to 18% to the conventional bottom-fixed off-shore power plants. When the technology is matured and shown to work beyond 1 MW that Vestas already tested, also the capital expenditures (CapEx), a solid €1.5 mln lower than at location 2, will be a
likely driver for changing to the MRS. A perfect country to give it a try is a developing economy, such as Poland, where at least 60% of the logistics supply chain for the off-shore industry is in place, but where the marine infrastructure does not allow to immediately compete with global single large rotor OEM producers.



## 3 Loads and Performance Estimation of MRS

A concept of a MRS wind turbine was born early in the 20th century (Jamieson, 2011), when the main factor limiting the size growth was steel used for blade manufacturing. Development of modern composite materials offering much better strength-to-weight ratios solved the problem for a while. Nowadays, wind turbines are up-scaled and again, the material weight and costs are starting to hit the limits, especially the economic ones. Hence, MRSs are reconsidered.

The essence of multi-rotor economics lies in maximizing the area to volume relationship of a rotor. Capacity of a large single rotor wind turbine may be obtained by a set of smaller rotors having the same swept area simultaneously decreasing the mass of rotor-nacelle assemblies (RNA). Moreover, at a distant sea location multi-rotor solution ensures the power generation continuity in the event of a rotor failure, a very plausible fault in harsh off-shore environments. Where the conventional single rotor turbine becomes non-operational, the MRS capacity decreases proportionally only to the number of faulty rotors. Maintenance actions do not have to be taken immediately and a number of extremely costly service expeditions is reduced. If aerodynamic loads are considered, a turbulent wind field will affect a multi-rotor system differently than in the case of a single large unit. In a single rotor it will produce greatly unbalanced loads along a blade making a fatigue failure more probable. Theoretically, in MRS these loads may be efficiently mitigated and redistributed by independently-controlled smaller rotors. Additionally, counter-rotating and symmetrically co-located in-plane turbine sections can notably diminish the net torque on the support structure by using differential thrust. Finally, MRS helps in the standardization of rotor and drive train components bringing unit costs down.

These MRS pros are partially offset by some problems, the first one being the complexity of the support structure needed for connecting the rotors together. It constitutes a meaningful fraction of the final mass and causes the dynamic behaviour of the system to be more unpredictable. Second of all, although technically capable of carrying the scaled down weight of MRS, the tower yaw bearing needs to be placed down below the bottom row of rotors and thus closer to the sea surface. Lastly, in the turbulent sheared inflow the arrayed turbines will eventually operate at slightly different speeds and non-uniform blade pitch angles. This may obliterate coherent loading on the support structure. However, since each rotor is much smaller relative to the main structure, all dynamic effects, such as tower passage, may be more benign than for a single rotor turbine.

### 3.1 Metocean conditions

To derive a conceptual deep off-shore MRS, a representative metocean state must be established. For this study the data was provided courtesy of German Institute of Coastal Research (ICR) that granted us the access to model-based data bank called coastDat1. Data came from a sequence of numerical models to reconstruct all aspects of marine climate across many decades relying only on large-scale atmospheric conditions or bathymetry. CoastDat1 has been used extensively in the process of designing, planning and installing off-shore wind farms, especially in Danish and German projects (Weisse et al., 2008).

We extracted wind directional speed components at 10 meters AMSL from an atmospheric hourly hindcast for the period 1958-2007. Only a spatial resolution of 50x50 km grid was available so a data point nearest to the location 4 was picked.

The records were used for calculation of the 50-year mean wind speed ($V_{ave50}$=10.08 ms$^{-1}$) and its maximum value ($V_{max50}$=36.70 ms$^{-1}$) at hub height of 90 m applying the power law with shear exponent $\alpha$=0.14. Figures 4 and 5 present the available wind data graphically. The directional behaviour was depicted by a wind rose graph (Pereira, 2020). The significant wave height $H_S$, wave directional behaviour and mean wave period $T_w$ were available for a slightly shorter period of time (1958-2002). The maximum value of the significant wave height $H_{Smax}$ and the corresponding wave period $T_{wHs}$ as well as mean parameters ($H_{Savg50}$ and $T_{wavg50}$) for the entire period were computed to be $H_{Smax50}$=9.89 m, $T_{wHs50}$=12.3 s, $H_{Savg50}$=1.20 m and $T_{wavg50}$=5.19 s.



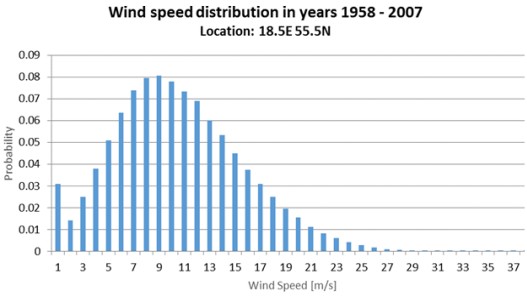

**Figure 4: Wind speed histogram for the location in the vicinity of the proposed MRS system (data source: ICR).**

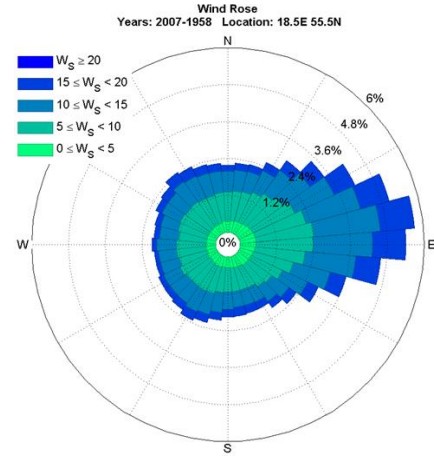

**Figure 5: Wind directional distribution for the selected grid point (data source: ICR).**

**3.2 MRS numerical model development**

In the next steps we show the path to the load analysis using aero-servo-elastic simulations prepared in FAST v.8.0 (Jonkman and Buhl, 2005). In order to focus on multiple rotor modelling, we assumed that regardless the hydro loads, the mooring system is able to completely take away all degrees of freedom (DOF) of the platform ensuring its perfectly inelastic

operation, thus simplifying the analysis significantly. It is obvious that for a detailed design the hydrodynamic effects need to be taken into account.. To make the floating MRS simulation in FAST possible, the following steps were undertaken:

1. Down-scaling of the baseline 5MW NREL wind turbine (Jonkman et al., 2009) to a 7x smaller individual single rotor of the system (1MRS) taking geometry, mass, aerodynamic performance, and control system into account.
2. Building a FAST model representing the 1MRS rotor and preparation of all required input files.

3. Development of a MATLAB script modifying the FAST functionality in order to save time and computational power during in-loop simulations.
4. Running FAST simulations in order to check the responsiveness of the 1MRS rotor against some of the load cases specified in (Germanischer Lloyd WindEnergie, 2005).
5. Analysis of blade tip deflections and blade root bending moments plus subsequent comparison with results obtained

for the baseline 5 MW NREL turbine.



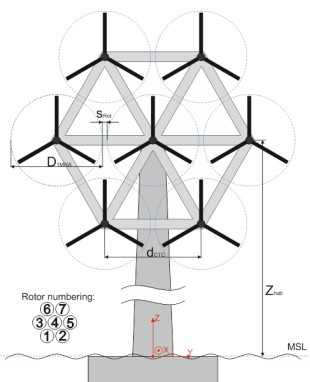

**Figure 6: Schematic visualization of the proposed MRS system (own depiction).**

The proposed MRS consists of 7 rotors of 714 kW each amounting to 5 MW of rated power in total (figure 6 and table 4). To make it possible we scaled the baseline 5 MW introduced by NREL to about 48 m in diameter. The geometrical centre of the

collective swept area (corresponding to COG of rotor number 4) for all 7 rotors is placed exactly at the hub height of the baseline wind turbine; rotors are symmetrically placed with respect to the tower yaw axis to minimize the imbalance of axial loads. Three out of seven rotors spin in the opposite direction to mitigate the tendency of the system to roll about the X axis. The maximum significant wave height of 9.8m served as means to validate the position of the bottom row of rotors, where a clearance of 20m between the lowest position of the rotor blade tips 1 and 2 and the MSL ensures operational safety. The

minimum spacing between blade tips for neighbouring rotors was assumed to be 5% of the 1MRS diameter. Therefore, the centre-to-centre distance between adjacent turbines is equal to ca. 50m.

**Table 4: Variation of 50-year mean speed $V_{ave-CD1}$ vs. rotor location.**

| Rotor no. | $Z_{hub}$ (m) | $V_{ave-CD1}$ (m/s) |
|-----------|---------------|---------------------|
| **6, 7**  | 133.3         | 10.65               |
| **3, 4, 5** | 90.0        | 10.08               |
| **1, 2**  | 46.7          | 9.20                |

Next, the scaled 1MRS turbine was iteratively tested in FAST until its Cp=f(TSR) curve compared to the full size baseline. The analysis included a study of blade and tower structural properties, defining wind profiles and scaling the airfoil lift/drag

characteristics as well as adjustments to the servo controller library. A time step study yielded an optimal value of 0.005s to ensure stable convergence while accounting for sudden changes in the wind speed. All DOF for blades (2 flap modes and 1 edge mode), generator, nacelle (a yaw bearing), and tower (2 fore-aft modes and 2 side-to-side modes) were obeyed. To conserve space the effect of mass reduction in table 5 is shown. For information on model scaling see (Anant et al., 2012).

**Table 5: Masses and moments of inertia obtained via geometric scaling.**

| Components | 5 MW (kg) | 1MRS (kg) | 7MRS (kg) |
|-----------|-----------|-----------|-----------|
| **Blades** | 53 220 | 2 874 | 20 120 |
| **Hub** | 56 780 | 3 066 | 21 460 |
| **Nacelle** | 240 000 | 12 960 | 90 710 |
| **Total RNA** | 350 000 | 18 900 | 132 300 |

**3.3 Study Cases for turbine loads**

Due to software limitations, the aerodynamic interaction between individual rotors of the MRS system could not be modelled. However, studies by (Smulders et al., 1984) and, more recently, (Ransom et al., 2010) did prove that adjacent



rotors suffer no major power loss when arrayed as close as 0.05D to each other. Similar observation was made in a CFD
study by (Chasapogiannis et al., 2014). A series of FAST simulations was prepared and carried out in order to assess the
responsiveness of both, the baseline 5 MW NREL wind turbine and the MRS system, to the weather conditions defined in
accordance with the Germanischer Lloyd WindEnergie (GL) guidelines (Germanischer Lloyd WindEnergie, 2005):

    1. Normal wind profile model (NWP) – at 8 m/s, 11.4 m/s, 18 m/s,

    2. Normal turbulence model (NTM) – avg. wind speed $V_{ave50}$ with medium Tu category $B$ ($I_{15} = 0.16$ and $a = 2$),

    3. Extreme wind speed model (EWM) – turbulent 50-year extreme wind speed $V_{ref} = 50$ m/s,

4. Extreme operating gust (EOG) – $V_{ave50}$ and 25ms$^{-1}$ steady wind with extreme gust speed $V_{gust50}$.

Prior to the design of the rotor support structure, the attention is put on another fairly sensitive part of a rotor: the blades. The
set of extreme load cases listed is likely to be the driving factor for the design of the rotor and the support structure as well. It
is required to assess the impact of extreme direction changes (EDC) as this load scenario is clearly different for a MRS.

Considering the extreme loading scenarios, we aimed for the minimum clearance between the blade tip and the tower to be
no less than 30% of the default value when the turbine is parked. We thus computed the net clearance of the baseline turbine
($NC_{NREL}$=10.5m) using the overhang distance, the shaft tilt line, the precone angle and the tower diameter from (Jonkman et
al., 2009). In absence of the definition for the support bars in the MRS, we used the scaled tower diameter of the baseline
turbine. The net clearance ($NC_{IMRS}$) for 1MRS rotor is estimated at 1.92m. If in any of the simulated scenarios the blade
flapwise deflection (out-of-plane or $Bl\_DefOoP$) exceeded these limits, it was considered that one of the turbine blades
crashed into the support member. The acceptable blade tip deflections are "safe" in an operational margin up to 70% of $NCs$,
that is <7.38 m for the baseline turbine or <1.34 m for the 1MRS.

### 3.3.1 Normal wind profile model

In this model, responsiveness of the system for a few steady wind speeds (8 m/s, 11.4 m/s and 18 m/s) was tested.
Simulations were run for 100 seconds while results from the last 30 were considered. During the first 70 seconds, if
necessary, the rotor blade pitch was allowed to be adjusted. Difference in the rotor locations (height) was taken into account
and wind speed was altered according to a power law to accommodate for the sheared inflow over the entire MRS area.

**Table 6: Summary of MRS loads for normal wind profile conditions.**

| $n$ | $Z_{hub}$ (m) | $V$ (m/s) | RotTorq (kNm) | RotThrust (kN) | Bl_DefOoP (m) | Bl_DefInP (m) | Bl_RootMx (kNm) | Bl_RootMy (kNm) |
|---|---|---|---|---|---|---|---|---|
| **1/2** | 47 | 7.30 | 97.5 | 44.66 | 0.38 | 0.04 | 30.53 | 242.5 |
| **3/4/5** | 90 | 8.00 | 117.2 | 53.69 | 0.45 | 0.05 | 37.95 | 290.8 |
| **6/7** | 133 | 8.45 | 130.9 | 59.94 | 0.50 | 0.05 | 41.93 | 323.9 |
| **1/2** | 47 | 10.40 | 198.1 | 90.76 | 0.74 | 0.08 | 64.08 | 485.6 |
| **3/4/5** | 90 | 11.40 | 231.8 | 99.14 | 0.81 | 0.10 | 75.15 | 530.5 |
| **6/7** | 133 | 12.05 | 231.8 | 85.70 | 0.69 | 0.11 | 73.67 | 459.8 |
| **1/2** | 47 | 16.42 | 231.8 | 56.37 | 0.35 | 0.10 | 47.67 | 282.8 |
| **3/4/5** | 90 | 18.00 | 231.8 | 51.61 | 0.29 | 0.09 | 71.49 | 248.7 |
| **6/7** | 133 | 19.02 | 231.8 | 49.19 | 0.24 | 0.08 | 72.07 | 230.1 |

Maximum magnitudes of produced rotor loads for NWP (thrust, torque, and blade root bending moment) as well as blade tip
deflections are presented in table 6 for the MRS system and in table 7 for the baseline 5 MW NREL. The shorter blades of
the MRS rotors are stiffer and deflections change only slightly regardless of the blade position (variation in the range of
0.01-0.03 m). The significantly longer 5 MW NREL's blades display a more sensitive behaviour azimuth-wise. The out-of-
plane deflection alters periodically and the amplitude of these fluctuations reaches up to 0.4 m. For both systems the



deflections are the biggest for the rated wind speed. It is the most strenuous NWP speed for the rotor, because at higher velocities the control system helps to alleviate the axial and bending loads via blade-pitch adjustments.

**Table 7: Summary of 5 MW NREL loads for normal wind profile conditions.**

| $V$ | RotTorq | RotThrust | Bl_DefOoP | Bl_DefInP | Bl_RootMx | Bl_RootMy |
|---|---|---|---|---|---|---|
| (m/s) | (kNm) | (kN) | (m) | (m) | (kNm) | (kNm) |
| **8.0** | 2039 | 495.7 | 3.10 – 3.36 | 0.80 | 4277 | 6065 |
| **11.4** | 4224 | 783.4 | 4.91 – 5.31 | 1.12 | 4956 | 9772 |
| **18.0** | 4213 | 459.7 | 1.62 – 1.97 | 1.10 | 4929 | 5195 |

**3.3.2 Normal turbulence model**

We carried out simulations using turbulent wind profile prepared in the TurbSim software. Wind profiles were generated according to IEC 61400-3. Medium turbulence intensity category B ($I_{15} = 0.16$ and $a = 3$) was selected and characteristic 50-year mean speed $V_{ave50}$ was set as a reference value at the hub height. For other MRS-relevant heights the power law was used to adjust the mean speed as seen in table 4.


Each rotor of the MRS system was simulated for 630 seconds with its own unique and randomly-generated wind profile. Time step 0.01s was set for subsequent data outputs and maximum absolute values of parameters were determined. The performance of the 5 MW NREL turbine was checked against the same wind conditions as the central 1MRS rotor (number 4 at $Z_{hub} = 90$m). For all runs, the first 30 seconds were excluded from analysis, because it was impossible to make the initial conditions uniform in terms of the rotor speed or blade-pitch angle (each wind profile generated randomly).


Maximum values of loads and blade tip deflections for each individual 1MRS rotor and baseline NREL turbine are presented in table 8. In neither case the permissible blade tip clearances were violated, however rotor number 7 was very close to the safety margin. It is worth highlighting that the blade root bending moments (BRBMs) are on average 16 up to 29 times higher for the baseline turbine. It makes its construction extremely challenging. Moreover, the blades of the baseline turbine are more susceptible to bending (right imprint in figure 7) thus being more prone to fatigue fracture than the shorter MRS blade.


**Table 8: Summary of the maximum loads and deflections for NTM with $V_{ave-CD1}$ wind speed.**

| n | $V_{max}$ | RotTorq | RotThrust | Bl_DefOoP | Bl_DefInP | Bl_RootMx | Bl_RootMy |
|---|---|---|---|---|---|---|---|
|  | (m/s) | (kNm) | (kN) | (m) | (m) | (kNm) | (kNm) |
| **1** | 14.31 | 209.3 | 121.2 | 0.97 | 0.12 | 186.9 | 684.2 |
| **2** | 14.62 | 204.7 | 118.0 | 0.95 | 0.13 | 180.7 | 651.3 |
| **3** | 15.21 | 258.1 | 150.4 | 1.09 | 0.16 | 197.4 | 817.2 |
| **4** | 15.28 | 237.6 | 138.6 | 1.04 | 0.14 | 182.3 | 702.7 |
| **5** | 15.09 | 244.8 | 159.4 | 1.17 | 0.13 | 187.3 | 801.1 |
| **6** | 16.34 | 257.5 | 145.1 | 1.16 | 0.17 | 192.5 | 784.5 |
| **7** | 15.65 | 258.2 | 161.9 | 1.32 | 0.18 | 221.5 | 892.1 |
| **NREL** | 15.28 | 4324 | 857.2 | 6.17 | 1.17 | 5376 | 11370 |

Interestingly enough, the ability of the MRS system to extract power at turbulent wind is more stable (left imprint in figure 7). The correlation between the results for the baseline turbine and the MRS shown is very satisfactory, but the MRS system is visibly less prone to sudden variations in rotor power. The effect might wither away when aerodynamic interaction between rotors is observed. This is supported by work carried out in the INNWIND.EU project (INNWIND.EU, 2015).






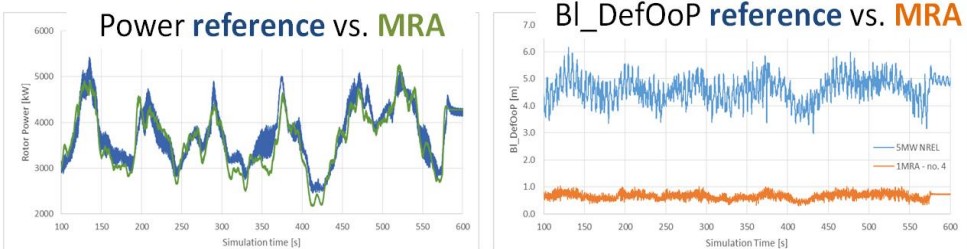

**Figure 7: Comparison of total rotor power and Bl_DefOoP variations between the baseline turbine and rotor no. 4, NTM with** $V_{ave-CD1}$ **wind speed.**

### 3.3.3 Extreme wind speed model

The analysis of meteorological data revealed that the 50-year maximum speed $Vmax_{50}$ is equal to 36.7 m/s ($Z_{hub}$). We decided, however, to evaluate the MRS system for the class I winds, where the extreme wind speed is $V_{ref} = 50ms^{-1}$ (table 9). This time rotors were tested with no correction for the sheared inflow. However, only MRS rotor no. 4 and NREL turbine used exactly the same wind input profile. The rest were randomly generated.

**Table 9: Maximum loads and deflections of MRS turbine for EWM with** $V_{ref}$ **wind profile.**

| n | $V_{max}$ (m/s) | RotTorq (kNm) | RotThrust (kN) | Bl_DefOoP (m) | Bl_DefInP (m) | Bl_RootMx (kNm) | Bl_RootMy (kNm) |
|---|---|---|---|---|---|---|---|
| **1** | 70.00 | 275.9 | 87.10 | 1.09 | 0.98 | 521.8 | 574.9 |
| **2** | 69.78 | 277.1 | 88.79 | 1.07 | 0.96 | 582.3 | 633.2 |
| **3** | 70.70 | 276.5 | 76.46 | 0.99 | 0.97 | 471.5 | 556.1 |
| **4** | 69.38 | 268.9 | 81.70 | 1.04 | 0.98 | 512.1 | 503.5 |
| **5** | 66.44 | 272.1 | 76.81 | 0.93 | 0.87 | 485.9 | 511.7 |
| **6** | 69.78 | 277.1 | 88.79 | 1.07 | 0.96 | 582.3 | 633.2 |
| **7** | 66.80 | 274.0 | 79.67 | 1.00 | 0.91 | 472.9 | 497.8 |
| **NREL** | 69.38 | 6854 | 809.2 | 6.10 | 5.87 | 11190 | 10590 |

Even though rotors in the MRS system experience much higher wind speeds than in the normal turbulence model, exerted thrust forces are smaller when compared to results from table 8. While a turbine operates in region 3, blade pitch is adjusted in order to reduce lift force acting on blades and to help in regulating the rotor speed. Unfortunately, the reaction of the control system is not instantaneous thus turbine structural response to significant changes in wind speed is not always sufficiently damped (bigger values of rotor torque). However, exuberantly higher inertia of the larger NREL blades prove to be even more problematic to control. In fact, normal operation in extreme turbulence is often a designing load case for major components of large wind turbines. This is not likely to be such a problem for the smaller MRS according to our results. The smaller rotors are more nimble and can react faster to changing wind conditions, another advantage of the MRS.

During extreme wind conditions, abrupt changes in magnitude and direction of velocity occur. The rotor size, its mass distribution and thus stiffness are crucial factors for containing the fluctuations. As shown in figure 8, the smaller MRS rotors prove to be a safer, more immune design.




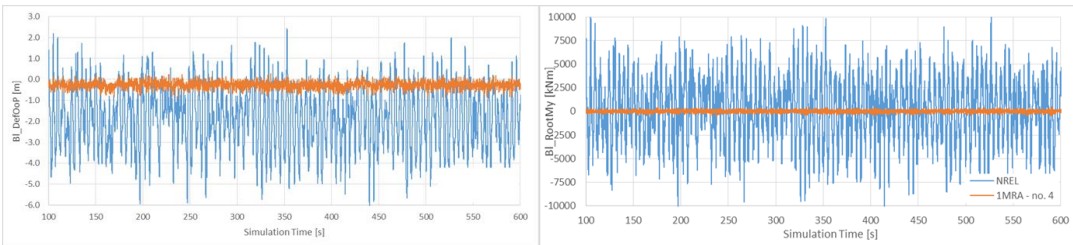

**Figure 8: Comparison of blade tip deflections and BRBMs about $Y_b$ axis during EWM for $V_{ref}$.**

It is also worth pointing out that in the EWM study the blade of the baseline turbine more frequently deflected in the direction of the oncoming wind (negative values in figure 8). It may be explained by the sudden changes in pressure distribution around the blade suction and pressure sides. As a result the blade moves in and out of its own wake.

### 3.3.4 Extreme operating gust

For analysis of the response of both turbines to the 50-year extreme operating gust at the hub height, we computed the gust magnitude $V_{gust50}$ for two instances of $V_{hub}$: the 50-year mean wind speed and the turbine cut-out speed of 25 ms$^{-1}$. After preliminary analysis, it was concluded that the case $V_{hub}=V_{ave50}$ is more hazardous. Thus, the estimate was $V_{gust50}$=11.01 ms$^{-1}$ with category A turbulence leading to the standard deviation of $\sigma_1$=2.11 ms$^{-1}$. We slightly modified the formula reported in (Germanischer Lloyd WindEnergie, 2005) to take a different starting time (20s after the simulation began) into account. Characteristic gust shape duration was $T_g=14$s as per (Germanischer Lloyd WindEnergie, 2005) with the total simulation time 120s. The results for the EOG model are presented in figures 9 and 10.

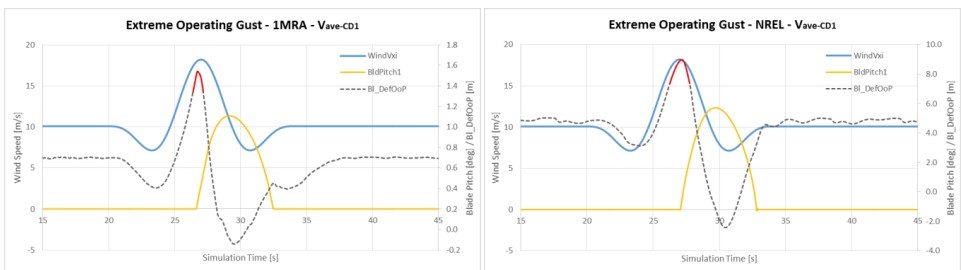

**Figure 9: Wind profile, blade pitch and out-of-plane blade deflections for EOG model: 1MRS (left), 5MW NREL baseline (right).**

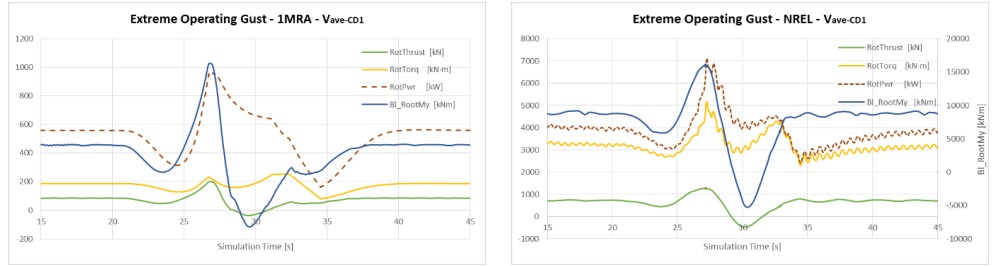

**Figure 10: Rotor thrust, torque and power as well as BRBM My, EOG model: 1MRS (left), 5MW NREL baseline (right).**

One of the biggest problems during the extreme operating gust is the delay in the controller reaction. Before the change in wind speed is recognized and appropriate counteraction taken, the blade is subjected to quickly increasing thrust force and deflects beyond the region of safe operation (sections highlighted in red). This means that the margin set as a maximum safe deflection was violated but the blade did not collide with the tower in either of the cases. This leads to the BRBM (about $Y_b$



axis) values higher than during EWM simulations, a potential fracture of the blade root. Within 3 seconds BRBM
magnitudes changed by around 1100 kNm and about 6500 kNm for the 1MRS and the NREL respectively (see figure 10).
Most of it was absorbed by the blade-hub connection. Rotor power production is slightly less sensitive here due to the inertia
and the mass of the rotor itself. Additionally, small oscillations of the blade after the gust has left the rotor area may be
observed.

**4 Conclusions**

Although a number of simplifications had to be made in order to modify the software for MRS systems and still yield
reliable results, the analysis is one of the few available on MRS systems and yields valuable first insights into the floating
MRS concept. After a thorough analysis and evaluation of the output parameters for all assumed models, a few conclusions
may be drawn (supported by data gathered in table 10). The nominal clearance NC between the blade tip and tower/truss
body defining the maximum flapwise deflection was never exceeded. Surprisingly the biggest values of *Bl_DefOoP* were
observed for the EOG conditions instead of EWM, where the wind speed magnitude occasionally reached stunning 70 m/s.
Testing for the extreme gust unveiled breach of the safety margin by 10.2% and 15.3% of the analysed designs respectively.

**Table 10: Summary of blade deflections and BRBMs for all investigated models.**

| Model | Bl_DefOoP | | | | Bl_RootMx (kNm) | | | Bl_RootMy (kNm) | | |
|---|---|---|---|---|---|---|---|---|---|---|
| | 1MRS | | NREL | | 1MRS | NREL | % | 1MRS | NREL | % |
| | (m) | NC% | (m) | NC% | | | | | | |
| **NWP$_{8.0}$** | 0.45 | *23.4* | 3.36 | *31.9* | 37.95 | 4277 | *-99%* | 290.8 | 6065 | *-95%* |
| **NWP$_{11.4}$** | 0.81 | *42.2* | 5.31 | *50.5* | 75.15 | 4956 | *-98%* | 530.5 | 9772 | *-95%* |
| **NWP$_{18.0}$** | 0.29 | *15.1* | 1.97 | *18.7* | 71.49 | 4929 | *-99%* | 248.7 | 5195 | *-95%* |
| **NTM** | 1.04 | *54.2* | 6.17 | *58.7* | 182.3 | 5376 | *-97%* | 702.7 | 11370 | *-94%* |
| **EWM** | 1.04 | *54.2* | 6.10 | *58.0* | 512.1 | 11190 | *-95%* | 503.5 | 10590 | *-95%* |
| **EOG** | 1.54 | *80.2* | 8.97 | *85.3* | 153.6 | 5056 | *-97%* | 1028 | 16190 | *-94%* |

As far as BRBMs are concerned, the downscaled MRS rotors experienced significantly lower stresses in the blade-hub
connection. The biggest reduction (up to 99% for 8 m/s steady wind) is observed in bending moments about $X_b$ axis where
mass of a blade is the crucial factor. According to the scaling model used, the mass of a single 1MRS blade would be 94.6%
smaller than the NREL blade, which is extremely important to sustaining mass-related loads such as moments of inertia.
Additionally, a longer blade experiences higher load imbalance along its span, but it also depends on current angular position
(azimuth angle). Higher velocities in the upper part of the rotor plane bring about greater loads. Combined with bigger rotor
radius, higher amplitudes of blade fluctuations are observed. That may lead to a premature fatigue failure. With the MRS
system this issue is partially solved, however appropriate construction of the support frame is necessary to take full
advantage of the underlying potential. Still a tentative support structure design is planned along with a separate study of
fatigue loads. The major difficulty there is the number of combined metocean conditions which should be considered.



## Appendices


-

## Code availability

The code used for the study can be made available upon a request.

## Data availability

Data sets used for the study can be made available upon a request.

## Video supplement

Not available.

## Author contribution

The paper was entirely written by Maciej Karczewski. Peter Jamieson supplied numerous input on the MRS technology,
Arnoldus van Wingerde as well as Bernhard Stoevesandt were consulted on the matters concerning the wind farm data
analysis, blade and rotor structural dynamics as well as aeromechanics, Piotr Domagalski and Lars Seatran contributed by
extending comments on the turbulent wind field data and resource-based assessment of a wind site potential. Additionally,
all of the authors have checked the document.

## Competing interests

The authors declare no competing interests.

## Acknowledgments

Authors are wishing to acknowledge assistance from students Sebastian Dziomdziora and Aleksandra Brygider, Lodz
University of Technology for their enormous help in gathering information leading up to this research. The many inputs on
COE computations provided by Mr. Pawel Mawduk from GSG Towers in Poland are much appreciated as well. Finally, a
separate thank you is extended to Vent LLC for a detailed list of wind turbine deployment costs in Poland.

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
