# Peer review of "Potential of load and O&M costs reductions of Multi Rotor System for the south Baltic Sea"

_Wind Energy Science, 2020_

## Referee Comment (RC1) · Anonymous Referee #1 · 31 May 2020

Referee's comments: This paper describes an analysis of the economic aspect, power output performance and wind load issues for a MRS composed of 7 rotors compared to a single bigger standard wind turbine of 5MW size. The analyses are reasonable, very interesting, innovative and useful, although there are a lot of assumptions for the analyses. It seems to the referee that the present paper should be acceptable to WES.

For minor points, there are a couple of questions and comments as follows:

1. Page 9, 3.3 Study cases. . . : It seems to the referee that the aerodynamic interaction between individual rotors is important. If all the rotors are operated normally under a small shear and a small fluctuation in wind speed and direction, the interference may be small. However, if each rotor in a MRS is operated with a large different rotation speed due to some causes, like one rotor failure or inherent fluid dynamic interference

[Figure]

for the flow around multiple bodies in a smooth or turbulent flows, the wind load for each rotor is different largely, it may give a significant irregularity and/or a bias in the power output and wind load to the MRS system. Please imagine a case that one rotor is operated and the other is stopped with one tower. The thrust force for each rotor is very different. What happen in the MRS under this situation? How do the authors think of it? 2. Page 10, table 6: What are the definitions of Bl_DefInP and BL_ DefOoP ? 3. Page 10, table 6: How those values are evaluated for Bl_ootMx and My? Just depend on the height only? 4. Page 11, 3.3.2 Normal turbulence model : Does the TurbSim include the fluctuations in the wind direction? 5. Page 11-13, in the Figures 7-10: The denotations for the symbols and lines are so small to read. 6. References: it should be written in more comprehensive way.

Please also note the supplement to this comment:
https://www.wind-energ-sci-discuss.net/wes-2020-23/wes-2020-23-RC1-supplement.pdf

---

## Referee Comment (RC2) · Peter Dalhoff (Referee) · 1 Jul 2020

The paper addresses two highly interesting and important topics - loads and O&M of Multirotor systems. Therefore it is seen as highly relevant.

The paper appears to be two papers, one on cost modelling (chapter 2) and one on loads (chapter 3), whilst the conclusions seem to address only loads and not costs.

Although it is stated, that the whole paper has been written by the main author, it reads as two papers from different writers.

Chapter two on costs reads more like a business report with many assumptions on input parameters and a favourable result for MRS. That chapter lacks a scientific and deep analysis on costs and on O&M for MRS. E.g. a simple assumption of availability

of 98% regardless of turbine type needs to be analysed. What is the failure rate and failure consequences of an MRS? How would the maintenance strategy look like and what is the impact on LCoE?

Chapter three deals with loads. Assumptions, input parameters, load cases are clearly stated. Results are clearly depicted. Discussion of the results is very interesting, but sometimes it seems a bit unfair comparison against the Single Rotor. E.g. it is obvious, that a small rotor produces smaller loads and deflections than a large rotor. Therefore fair comparisons need to be made, such as accumulated thrust forces or deflections and moments normalised on their mean value.

The referee recomends this paper to be published with major revisions in chapter 2, minor revisions in chapter 3 and a conclusion, which addresses all findings.

See pdf also

Please also note the supplement to this comment:
https://wes.copernicus.org/preprints/wes-2020-23/wes-2020-23-RC2-supplement.pdf

**Supplement:**

[revised manuscript text omitted]

---

## Author Comment (AC1) · 7 Aug 2020

Dear Reviewer,

We thank Anonymous Referee 1 for the time and comments. In the following, we will do our best to reply to the comments and propose improvements for the final manuscript.

Enclosed please find the changes made to the original manuscript following the review by Anonymous Referee.

Anonymous Referee minor remark #1: Page 9, 3.3 Study cases… : It seems to the referee that the aerodynamic interaction between individual rotors is important. If all the rotors are operated normally under a small shear and a small fluctuation in wind speed and direction, the interference may be small. However, if each rotor in a MRS is operated with a large different rotation speed due to some causes, like one rotor failure or inherent fluid dynamic interference for the flow around multiple bodies in a smooth or turbulent flows, the wind load for each rotor is different largely, it may give a significant irregularity and/or a bias in the power output and wind load to the MRS system. Please imagine a case that one rotor is operated and the other is stopped with one tower. The thrust force for each rotor is very different. What happen in the MRS under this situation? How do the authors think of it?

The authors' reply to Anonymous Referee minor remark #1: The referee is right pointing a single major failure of a rotor enclosed in the MRS form. The problem of control system for MRS is very profound and has been under investigation by many researcher, including one of the co-authors of this article, Peter Jamieson. Article "Yaw Control for 20MW Offshore Multi Rotor System" by Euan MacMahon et al., in: European Wind Energy Association Annual Event (EWEA 2015), 2015/11/17-20, Paris, points that a novel yawing technique applied to adjust the thrust of the individual rotors is very feasible. For example, if a clockwise motion is required to reduce the yaw error, the thrust of the rotors on the right hand side of the yaw axis, may be reduced. If a failure happens on the other hand, same system can be used to alleviate the increased structure loads. The article goes on to explain that aside a standard FEC (Full Envelope Controller) that is used to maximise power output below rated wind speeds, keep the power production constant above rated winds, reduce loads on a turbine, another control device can be used – the PAC (Power Adjusting Controller). With feed forward controller the PAC, in relatively short amount of time as it has been simulated, can be used to alter the power output of the wind turbine thus causing a change in the thrust force acting on the rotors, what translates to inducing a yaw moment on the tower. This can be used to counter the effects on single rotor failure. Of course the described control strategy is just highlighted here and would require optimizing for minimum power loss and various failure scenarios in mind.

Anonymous Referee minor remark #2: Page 10, table 6: What are the definitions of Bl_DefInP and BL_ DefOoP ?

The authors' reply to Anonymous Referee major remark #2:  The parameters *BL_DefInP* and *BL_DefOoP* are blade deflection in-plane and blade deflection out-of-plane, respectively.

Anonymous Referee minor remark #3: Page 10, table 6: How those values are evaluated for Bl_ootMx and My? Just depend on the height only?

The authors' reply to Anonymous Referee major remark #3: The notations *Bl_RootMx* and *Bl_RootMy* refer to the 'edgewise' (in $Y_bZ_b$ plane) and 'flapwise' (in $X_bZ_b$ plane) blade root bending moments respectively. The local blade-oriented rotating coordinate system $X_bY_bZ_b$ is shown in the figure below.

The blade root bending moments are based on fully discretized blade geometry, section by section respecting local chord, radius and force.

[Figure]

Fig. A11. Schematic representation of wind turbine rotor and nacelle assembly plus tower; NC - nominal clearance between tower and blade tip (own picture)

Anonymous Referee minor remark #4: Page 11, 3.3.2 Normal turbulence model : Does the TurbSim include the fluctuations in the wind direction?

The authors' reply to Anonymous Referee major remark #4: FAST simulations were carried out using turbulent wind profile prepared in the TurbSim software. It is a full-field, turbulent-wind simulator allowing a user to create wind profiles according to the specific IEC standard (in this case the IEC 61400-3 which is the version suitable for offshore wind turbines). Medium turbulence intensity category $B$ ($I_{15}$ = 0.16 and $a$ = 3) was selected and characteristic mean speed over the period of 50 years $V_{ave-CD1}$ was set as a reference value for the hub height of 90 meters. The wind speed fluctuations were stochastically simulated in each time step.

Anonymous Referee minor remark #5: Page 11-13, in the Figures 7-10: The denotations for the symbols and lines are so small to read.

The authors' reply to Anonymous Referee major remark #5: The aforementioned figures have been enlarged.

Anonymous Referee minor remark #6: References: it should be written in more comprehensive way.

The authors' reply to Anonymous Referee major remark #6: Where it was possible, the authors made improvements and added more details to cited works.

Sincerely Yours,

---

## Author Comment (AC2) · 7 Aug 2020

Dear prof. Dalhoff,

We thank you for the time and comments. In the lines that follow, we will do our best to reply to the comments and propose improvements for the final manuscript. At first, we focus on the comments that were posted in the text of the manuscript. Secondly, the general comments from the cover letter will be answered.

Reply to Reviewer's comment #1, pg. 1, line 18: The 5MW baseline turbine was selected as a well described, tried and tested model for this turbine was available to the authors. The described 5MW offshore turbine model, its development and accuracy has been tested by numerous researchers, the article:

Jonkman J., Butterfield S., Musial W., and Scott G.: Definition of a 5-MW Reference Wind Turbine for Offshore System Development, Technical Report NREL/TP-500-38060 February 2009

describes that in detail. With comparison of newer MRS in mind, the authors decided to use a tried and tested generic model of 5 MW that is, in turn, based on real Repower turbine. It is without a doubt that making a similar study for a turbine of size 14 of even 20 MW would have been more timely, however, it would introduce an additional variable: a model of a single large rotor wind turbine to which no data exists in order to verify it.

Reply to Reviewer's comment #2, pg. 1, line 19: The focus has been given to a rotor alone for few reasons. First of all, the authors wanted to confirm the degree of the reported reduction of loads in MRS in comparison to large single offshore turbines. This has not been done for any specific site on Polish territorial waters, where the input wind and wave data is available. Second of all, in the process of creating the legislation, know-how and logistics chain in Poland for erecting offshore wind turbines, it is important to consider alternatives to large single rotor turbines early on. From the economic study presented in the paper, it was made evident that MRS can be a viable way to cut costs in Capex and Opex financing stages. If proved mechanically sound, many existing smaller rotor manufacturing and transportation techniques could have been used instead of employing specialized vessels and ships on top of importing foreign technologies. Finally, the gangways of major ports in Poland, including Gdynia shipyard, which was selected as the installation and service port for Polish offshore programme, are deep enough to allow the construction of lighter MRS in-house with later towing to the operational site. Unfortunately, this is not possible with single large rotor turbines.

Reply to Reviewer's comment #3, pg. 1, line 22: We changed the word to "minimization".

Reply to Reviewer's comment #4, pg. 2, line 52: The sentence was changed to a passive one.

Reply to Reviewer's comment #5, pg. 2, line 64: Indeed, it is known that the real offshore wind shear is higher, (even from the authors' own work: https://doi.org/10.1115/1.4041816, https://doi.org/10.5194/wes-5-391-2020 ) but the main purpose of the presented paper was to show the calculations according to the written standards to provide rigid, consistent and universal calculations base.

As the calculations were based upon the standardized load cases, authors decided to keep the same type of source for the environmental data. Authors agree with the Reviewer that the data source could be updated i.e. taken from DNVGL-RP-C205 Environmental conditions and environmental loads, although even here, the recommended shear exponent is bigger than reported in the field (according to the literature).

Reply to Reviewer's comment #6, pg. 3, line 75: Yes indeed, the graph shows the capacity factors (CF) not Annual Energy Production (AEP) figures. It was corrected.

Reply to Reviewer's comment #7, pg. 3, line 72: The original data set contained only the production (AEP) figures, year by year. The post-processing thus required the authors to compute the yearly capacity factors, average the turbine output over the years and arrive at a single CF value.

Reply to Reviewer's comment #8, pg. 3, line 84: The authors do not understand this comment. The data comes from Danish Energy Agency and presents the actual MWh hours produced by each turbine on-site. Thus the presented capacity factors are computed by weighing the MWh actually produced by the turbine nameplate capacity. This definition of CF is widely used for example in NREL reports in order to judge the LCOE (Levelized Cost of Energy) of offshore wind projects.

Reply to Reviewer's comment #9, pg. 4, line 98: The rate given is "per operation" rather than "per day".

Reply to Reviewer's comment #10, pg. 6, line 147 in regards to turbine availability: The MRS availability is likely to be higher than that of a single rotor turbine. When one turbine in MRS array malfunctions, others are able to maintain production rate. In this scenario, the stated assumption is rather conservative. If however, all turbines in MRS array malfunction, then the availability rate is likely the same as in case of a single large rotor.

Reply to Reviewer's comment #11, pg. 6, line 147 in regards to LCOE: The definitions used in the source from 2005 were adjusted for inflation based on economic factors taken from the National Bank of Poland in regards to US dollar. The fixed charged rate was 11.58% and was used across all the sites. In comparison to a newer version of the same model (i.e. Stehly, T., Heimiller, D., Scott, G.: 2016 Cost of Wind Energy Review, National Renewable Energy Laboratory, Technical Report NREL/TP-6A20-70363, December 2017) where a value of 7.9% is reported, this approach presents as more conservative.

The remaining input parameters were as follows:

- Turbine Capital Cost (TCC): machine with transformer, marinization
- Balance of Station Cost (BOC): Transportation, assembly, RNA and erection, grid connection expertise cost and cost of obtaining the connection permits, location and environmental permits costs, geological meas. and planning permit costs, costs of energy, project and construction permits, here also we included port and staging costs, foundation, generator, materials costs, costs of engineering personnel: construction, electrical, building commissioning, plus equipment costs and scour protection
- Financial costs included: insurance, construction financing, decommissioning bond, and contingency expenditures
- Annual Operation costs: sea bottom lease, accounting, servicing, administration, telecommunication, own energy costs.

Reply to Reviewer's comment #12, pg. 6, in regards to Operation and Maintenance strategy of MRS: The reviewer is right that:

- a comprehensive look at service/maintenance strategy of MRS,

- the study of consequences of failure rates of MRS in comparison to a single large offshore turbine,
- the analysis of MRS availability scenarios in relation to well-known rates of single rotor turbines,
- the derivations on the number of service teams and vessels for an open-sea MRS operations

should all be dealt with. However, the data on availability of MRS and likewise its failures rates (i.e. from Vestas multi-rotor tests) is not made public. Moreover, although considered jointly with wind speed, the primary factor to determining turbine downtime is always(!) wave height, but also wave period, wave direction, and current. For all of the sites, this would require a formidable task of assembling open sea wind-wave maps and not only localized ones for sites 2/3/4. Instead, the authors assumed that availability rate is similar as for single rotor turbines, what, as it was explained, presents a rather conservative approach.

Furthermore, the maintenance strategy and number of service teams employed be it in the formula "tow-to-port" (for a floating turbine) or in-situ repair via SOV and/or MOV stationed on open sea or mobilized from port are all major focus of on-going research works aimed at optimizing O&M cost structure. This sort of analyses is much easier when prepared for the developed offshore markets such as Denmark, Germany or Great Britain and when conducted for single rotor turbines. To the authors best knowledge, there is little to no information in public domain on comprehensive number-based O&M strategy analysis in regards to MRS maintenance: bottom-fixed or floating. Therefore, it is even a more formidable task to prepare such study for Polish territorial waters, for which little has been analysed for single rotor turbines, much less to say for MRS.

The goal of the study presented in the paper, was to give a rather general view of the O&M cost aspects of single and MRS turbines in the market that is being formed. With this purpose in mind, the data used to derive the discussion about possible impact on O&M cost structure was based on standards set in InnWind project, in which one of the co-authors, Peter Jamieson, was involved. A number of other studies cited in the paper helps to decipher how did the authors arrive at the values that were used to build the cost model as presented.

Reply to Reviewer's comment #13, pg. 7 line 193: The authors are aware of this solution that was actually presented by the reviewer during one of the MRS workshops.

Reply to Reviewer's comment #12, pg. 8 in regards to comparing loads of MRS and a single rotor large turbine:

The required sentence was inserted in the newest version of the manuscript, line 271.

Reply to Reviewer's comment #13, pg. 11 line 292: The wind fields were not continuous in space, but indeed randomly created for a mean wind speed (see for example column no. 3 V(m/s) in Table 6) that was derived from the assumed wind shear profile at the respective hub heights of all seven rotors.

Reply to Reviewer's comment #14, pg. 11 line 300: The authors are not sure what is understood under the words "basic scaling". The authors are not sure what is meant under the phrase "the big rotor has to fight rotational sampling of turbulence".

Perhaps the answer to the reviewer's question about the scaling would be the explanation on how the authors scaled the model turbine. The Froude scaling law was used to scale the mass, moments of inertia, mass per unit length as well as material stiffness ratio. The MRS blade model was made to exhibit 1:1 aeroelastic similitude to its full-scale counterpart. That was achieved by matching non-dimensional parameters that govern the aerodynamics,

the structure and their coupling. The geometrical similarity in terms of shape and orientation of all components was maintained. Matching of Mach and Reynolds number is required, however investigated wind speeds make the first condition negligible. Unfortunately, one of the biggest weaknesses of the Froude model is inconsistency in scaling of Reynolds number. Reducing scale usually means entering into laminar-turbulent transition phenomena. It leads to increased importance of friction forces on the surface of a blade influencing pressure distribution on the model's rotor. This negative impact was partially compensated by keeping the parameters that normally would depend on this number (lift/drag coefficients for blades) unchanged.

From the structural force-deflection point of view, similitude required that the ratio of stiffness to aerodynamic forces be maintained as well as the distribution of the stiffness throughout the structure. The relevance of vibration amplitudes at reduced scales imposes mass ratio and stiffness ratio to be maintained. Additionally, in order to ensure consistent rotational characteristics, design TSR was preserved.

Reply to Reviewer's comment #15 pg. 12 left imprint of Fig. 7: Yes, MRA is MRS. The reviewer is advised that the plot presents the comparison of instantaneous wind turbine power between MRS, numbers of single rotors added together to create a total, and a single large rotor. The energy production rate for the portion visible on the plot is 559 kWh for MRS and 573 kWh for the large rotor.

Reply to Reviewer's comment #16 pg. 12 right imprint of Fig. 7: The normalised amplitudes for the tip deflection of the single large rotor and the no. 4 rotor of the MRS are presented below.

[Figure]

[Figure]

Reply to Reviewer's comment #17 pg. 12 line 315: The wind drag on the support structure was not considered. As stated in the text, the model entailed the studies of rotors and blades only.

Reply to Reviewer's comment #18 pg. 13 in regards to imprints of Fig. 8: The authors did not debate on the fairness of the approach taken. Perhaps the most vivid example to show the advantages of MRS in relation to a large rotor in such study would be to present an accumulated effect of all load cases presented on a fatiguing stress and the ability of both rotors (large and small) to cumulatively withstand such load combinations. However, it then would require extremely strenuous calculations in spectral domain of amplitude vs. frequency (i.e. Kaimal) instead of amplitude vs. time. A rainflow counted loads would then present some accumulated impact over time, but these were not the goals here. It would be an interesting question to answer in another study.

Reply to Reviewer's comment #19 pg. 13 in regards to imprints of Fig. 9: The extreme direction change was not made part of this study, unfortunately.

The following section aims at addressing the major comments that the Reviewer has made in the cover letter.

1) The reviewer writes "The paper appears to be two papers, one on cost modelling (chapter 2) and one on loads (chapter 3), whilst the conclusions seem to address only loads and not costs. Although it is stated, that the whole paper has been written by the main author, it reads as two papers from different writers."

Reply to Reviewer's major comment #1 from the cover letter: The paper was indeed written by Maciej Karczewski and supplemented by vast comments and inputs from all co-authors. In fact, the paper was written in 2 stages. At first the section about the loads was created, and then, after certain amount of time, the second section on O&M analysis was added. Perhaps this is why the paper "reads" as two separate papers. Finally, the input from co-authors was also spread over time and could introduce this effect of "duality" in the way the text sounds. Hopefully, this answers the referee's concerns about the authorship.

2) The reviewer states "Chapter two on costs reads more like a business report with many assumptions on input parameters and a favourable result for MRS. That chapter lacks a scientific and deep analysis on costs and on O&M for MRS. E.g. a simple assumption of availability of 98%

regardless of turbine type needs to be analysed. What is the failure rate and failure consequences of an MRS? How would the maintenance strategy look like and what is the impact on LCoE?"

Reply to Reviewer's major comment #2 from the cover letter: The authors believe that the incentives behind the O&M model that were explained above will satisfactorily answer the questions that the referee posted. Without a doubt a comprehensive MRS O&M model with deep analysis on costs and O&M for MRS would only lifted the scientific content of the very section in question. However, the authors feel that the number of unknowns is still too many for MRS O&M studies. Some background has been laid out by the aforementioned InnWind project and the findings made public there, plus more, have been used to structure the cost model for MRS in here.

Generally OpEx can vary greatly between projects for a number of reasons but the 3 largest cost drivers are:

- the meteorological ocean climate at the site (wind conditions and sea conditions: significant wave height, wave period, wave direction, and current);
- the distance from the wind farm to the maintenance facilities;
- turbine size.

The 3 are a direct consequence of the definition of O&M cost based per MWh or per MW or per kW of turbine nameplate capacity. The three cost drivers translate to turbine availability and thus further to its productivity rate counted in AEP [MWh/yr].

For the first driver, based on the sea/metocean conditions one can name a site to be:

- mild site
- moderate site
- severe site

on a scale proposed by US NREL. Each site where a floating turbine would be moored (bathymetry issues) is a severe site. Given this alone plus the weather conditions, one can safely say that the site no. 4 proposed in this document is a severe site.

For the second driver, the US NREL and Dutch ECN (Ashish Dewan, Masoud Asgarpour, "Reference O&M Concepts for Near and Far Offshore Wind Farms", December 2016, report no. ECN-E—16-055) recommend 4 maintenance strategies, based on site's proximity to service ports, best summarized in the table below:

**Table C-22. Matrix of O&M Strategies**

| | Close to Shore | Close to Shore+ | Medium Distance | Far Shore |
|---|---|---|---|---|
| **Alias** | **CS** | **CS+** | **MD** | **FS** |
| Description | Standard port-based O&M strategy | Standard port-based O&M strategy | Enhanced port-based O&M strategy | Mothership-based O&M strategy |
| Principle Access Vessel | Basic CTV[a] | Advanced CTV | SES[b] | CTV with mothership support |
| Wind Limit (m/s) | 20 | 20 | 20 | 20 |
| Hs Limit (m) | 1.5 | 2.3 | 2.5 | 2.5 |
| Vessel Speed (kn) | 20 | 20 | 35 | 20 |
| Access Vessel Day Rate | $2,800[c] | $6,500[c] | $9,000[c] | $2,800[c] |
| Passengers (#) | 12 | 12 | 12 | 12 |
| Shift Length (h) | 12 | 12 | 12 | 12 |
| Docking and Transfer Time (h) | 0.5 | 0.5 | 0.5 | 1.0 |
| Fuel Consumption Rate (gal/h): | 25 | 25 | 20 | 25 |
| Fixed Annual Maintenance Costs | n/a | n/a | n/a | $18,000,000[d] |
| Capital Investment | n/a | n/a | n/a | n/a |

[a] Crew transfer vessel

[b] Surface effect ship

[c] ECN User Guide.

[d] Communications with offshore wind industry.

Note: All other O&M equipment assumptions are from offshore wind industry feedback and NREL's offshore wind database.

Figure A. Source: "A Spatial-Economic Cost Reduction Pathway Analysis for U.S. Offshore Wind Energy Development from 2015–2030", Philipp Beiter et al., National Renewable Energy Laboratory, Technical Report NREL/TP-6A20-66579 September 2016

In practicality, any transfer time above 2hr to an offshore site is considered usually a far shore project. So combining the site class and distance to ports, one may derive O&M strategy and costs, for example as below (2 figures A and B from the above NREL report are provided). The plots are made for quay-side maintenance performed in port, to which the floating turbine is towed rather than in-situ repair. ECN recommends that for heavier replacements (spare parts above >3 MT). The regular corrective and condition based O&M is performed by a small service operating vessel (SOV) or a mother operating vessel (MOV), but it may largely depend on conditions at sea around turbine site but also en-route to the location.

[Figure]

**Figure 37. Availability results for the semisubmersible substructure**

Note: Mild, moderate, and severe sites are denoted in km.

[Figure]

**Figure 36. OpEx results for the semisubmersible substructure**

Note: Mild, moderate, and severe sites are denoted in km.

Figure B. OpEx results for the semi-sub floating turbine, the amounts are based for a generic 3.4 MW floating offshore unit -> Opex/kW is 100 mln USD/3.4 MW = 29.41 USD/kW

The above USD/kW estimates are a bit underrated, especially for a still relatively new concept – a single rotor floating wind turbine, much more in regards to placing a floating MRS. For the financial model that was used in the paper, to price O&M of an innovative floating turbine for Baltic Sea, the authors arrived at O&M of 77 EUR/kW, over 2.5 times more than reported in the strategy proposed by NREL.

3) The reviewer makes on opinion: "Chapter three deals with loads. Assumptions, input parameters, load cases are clearly stated. Results are clearly depicted. Discussion of the results is very interesting, but sometimes it seems a bit unfair comparison against the Single Rotor. E.g. it is obvious, that a small rotor produces smaller loads and deflections than a large rotor. Therefore fair comparisons need to be made, such as accumulated thrust forces or deflections and moments normalised on their mean value.

Reply to Reviewer's major comment #3 from the cover letter: The authors have provided two figures with normalised data. We thank the referee for making this point. It is also our observation that accumulated loads are going to penalize the rotor units of MRS in lesser degree than a single large rotor. The degree of this "penalty" could be rightly judged by quantitative and qualitative comparisons. We believe that quantitative comparisons would reveal full picture only if tallied in a form of i.e. rainflow counted ensemble of loads. But, as it was hopefully explained, this requires yet another model to be written on top of many assumptions already made for this study. Instead, the model as such is ready to be used for qualitative comparisons, yet with numbers represented in terms of absolute values in mind.

Sincerely Yours,

Maciej Karczewski